# Cold Acclimation and Supercooling Capacity of *Agasicles hygrophila* Adults

**DOI:** 10.3390/insects14010058

**Published:** 2023-01-06

**Authors:** Yiming Pei, Jisu Jin, Qiang Wu, Xiaocui Liang, Chen Lv, Jianying Guo

**Affiliations:** 1State Key Laboratory for Biology of Plant Diseases and Insect Pests, Institute of Plant Protection, Chinese Academy of Agricultural Sciences, Beijing 100193, China; 2Agricultural Genomics Institute at Shenzhen, Chinese Academy of Agricultural Sciences, Shenzhen 518100, China

**Keywords:** cold acclimation, supercooling, biological control, *Agasicles hygrophila*, *Alternanthera philoxeroides*

## Abstract

**Simple Summary:**

The chrysomelid beetle *Agasicles hygrophila* is an effective natural enemy used in the biocontrol of the invasive weed *Alternanthera philoxeroides.* However, few studies have focused on the cold acclimation of *A. hygrophila.* We investigated the effects of exposure to different sub-lethal low temperatures for different durations on biological indices of adult male and female *A. hygrophila* and subsequently selected an appropriate acclimation temperature to enhance low-temperature tolerance. We found that *A. hygrophila* can acclimate to the cold and that this acclimation can enhance the cold tolerance of this beetle at −10 °C. Moreover, an appropriate acclimation temperature and time were found to enhance the survival and longevity of captive-bred *A. hygrophila* adults, which has ecological relevance for the biological control of *A. philoxeroides* in cold areas of northern China.

**Abstract:**

*Agasicles hygrophila* Selman and Vogt is used in the biological control of the invasive weed *Alternanthera philoxeroides* (Mart.) Griseb. However, with the northward establishment of *A. philoxeroides* in China, the weak adaptivity of *A. hygrophila* to cold weather has resulted in the ineffective control of *A. philoxeroides* in northern China. Cold acclimation can significantly enhance insect cold tolerance, enabling them to cope with more frequent climate fluctuations. To improve the biological control efficacy of *A. hygrophila* in cold climates, we compared the effects of rapid cold hardening and acclimation on *A. hygrophila* under laboratory conditions. On initially transferring adults from 26 to −10 °C for 2 h, mortality reached 80%. However, when pre-exposed to 0 °C for 2 h and then transferred to −10 °C for 2 h, adult mortality was reduced to 36.67%. These findings indicate that cold acclimation can enhance the cold tolerance of *A. hygrophila* under laboratory conditions. However, the beneficial cold acclimation effects waned after more than 15 min of recovery at 26 °C. Exposure to 15 °C for 24 h or gradual cooling from 0 to −10 °C at 1 °C·min^−1^ also induced cold acclimation, indicating that long-term cold and fluctuating cold acclimation are also potentially effective strategies for enhancing low-temperature tolerance.

## 1. Introduction

Rapid cold hardening (RCH), a transitory adaptation to an extremely cold temperature that follows brief exposure at sub-lethal temperatures [1], is commonly observed in insects in the orders Coleoptera, Diptera, and Hemiptera, as well as in mites [2,3]. The hardening could have included the production of Heat Shock Proteins and cold-resistant substances [1]. The life cycles of many insect species include an overwintering stage, and insects that have recently passed through this stage tend to have greater cold resistance than before or at a more prolonged time after overwintering [4]. The ability of insects to endure brief spells of rapid cold acclimation can increase their survival under markedly changeable low-temperature conditions, particularly in spring and autumn when temperatures are liable to undergo considerable fluctuations or sharp declines [5]. Insects typically show plasticity with respect to low-temperature tolerance, which is essential for ensuring the growth, development, and reproduction of the insects in regions in which low temperatures are likely to be encountered. Indeed, phenotypic plasticity may account for the differences in the low-temperature tolerance shown by different geographical populations of the same insect species and can be considered to reflect the adaptive evolution of insects living in a given environment for a prolonged period [6]. For example, it has been demonstrated that when *Drosophila melanogaster* from temperate and tropical regions are kept at different temperatures for several generations, rearing temperatures determine changes in the cold tolerance of flies, rather than those of the collection sites [7].

Cold acclimation has been observed in both diapausing and non-diapausing females of the predatory mite *Euseius finlandicus* [8], and rapid cold acclimation has also been reported in numerous insect species, including *Cryptolestes ferrugineus* (Stephens), *Oryzaephilus surinamensis* (L.), *Rhyzopertha dominica* (F.), *Sitophilus oryzae* (L.), *Corythucha ciliata* (Say), *Euseius finlandicus*, and *Tribolium castaneum* (Herbst) [8,9,10]. Although these insects may overwinter at all stages, winter diapause has not been observed [11]. However, newly emerged *R. dominica* adults are more resistant to long-term low temperatures than the larvae [12]. Similarly, among the different stages of *C. ferrugineus*, adults are characterized by the highest cold tolerance [13]. These findings thus indicate that in several insect species, the adults may have cold acclimation ability and that the cold tolerance of adults may be notably higher than that of the pre-adult stages [12,13]. Important considerations with respect to insect cold tolerance are the supercooling point (SCP) and freezing point (FP), which respectively refer to the temperatures at which insect body fluids begin to freeze and eventually freeze [14]. The SCP and FP of insects are influenced by multiple endogenous and exogenous environmental factors, including developmental stage, age, body size, weight, rearing conditions, and external temperature, and are often used as indicators of insect cold tolerance [15,16]. 

*Alternanthera philoxeroides* (Mart.) Griseb, commonly referred to as alligator weed, is an amphibious plant species in the family Amaranthaceae that is native to South America [17], although it was introduced into suburban Shanghai, China, from Japan as a forage crop in the late 1930s, and subsequently spread to eastern China in the 1950s and southern China in the 1960s and 1970s, mediated by human activities [18,19]. However, the high reproduction rate of this plant has led to its rampant growth and spread, which has had adverse effects on agriculture, forestry, animal husbandry, and fisheries in those areas in which it has colonized and become established [20]. *Agasicles hygrophila* (Coleoptera: Chrysomelidae) was first introduced into China from Florida to control alligator weed in 1986 and has since been successfully used in this regard in southern China [21,22]. Temperature is an important factor influencing the growth, development, behavior, and evolutionary pathway of *A. hygrophila* [23], with low temperature having particularly pronounced effects on the occurrence, distribution, reproduction, and dispersal of this beetle. Recently, with the northward spread and establishment of *A. philoxeroides* in China, it has been established that *A. hygrophila* has a weak adaptation to cold weather and has therefore proved ineffective in the control of *A. philoxeroides* in these recently colonized northern regions. Accordingly, it is reasoned that enhancing the cold tolerance of *A. hygrophila* could make an important contribution to extending its distribution and establishment in northern China, and thereby improve its biocontrol efficacy.

In the subtropical and temperate regions of China, temperatures often drop below freezing during winter, thereby threatening the survival and successful overwintering of insects [24]. To prevent physiological or even fatal damage caused by low-temperature stress, insects have developed a series of physiological mechanisms to protect against the adverse effects of low temperatures [25]. Cold acclimation and the duration of the winter diapause are determined by a combination of cooling and changes in photoperiod during the periods from autumn to winter and winter to spring. Notably, however, insects can also rapidly acclimate to the cold in response to brief exposures (a few minutes) to low temperatures, which can protect against cold shock damage and thus enhance survival [2,26]. 

In this study, we compared the rapid cold acclimation ability of *A. hygrophila* adults, and we estimated the SCP of all stages to determine suitable insect statuses for cold acclimation and estimated the sublethal low temperature range. We compared and evaluated the rapid cold-hardening ability of *A. hygrophila* adults. To determine a standard cold shock exposure (identification temperature), we estimated the effect of rapid cold acclimation on the 80% survival rate. To quantitatively compare the effects of cold acclimation, we describe the cold acclimation reaction of this species. We also examined the persistence of rapidly cold hardening, according to the effective cold acclamation temperature and time, and determined the longevity of treated insects after cold acclimation. Finally, we discuss the ecological consequences of cold acclamation in terms of development, longevity, and biological control benefits.

The specific objectives of this study were to (1) determine whether *A. hygrophila* adults have cold acclamation ability; (2) establish the SCP and FP for the different stages of *A. hygrophila*; (3) identify suitable cold acclimation temperatures and times; and (4) assess the influence of cold acclamation on beetle longevity.

## 2. Materials and Methods

### 2.1. Experimental Host Plants and Insects

We planted *A. philoxeroides* in a greenhouse at the Institute of Plant Protection (IPP) of the Chinese Academy of Agricultural Sciences (CAAS), Haidian District, Beijing, China. The seedlings were subsequently transplanted into plastic pots (45 × 35 × 35 cm) containing a humus/soil mix at a density of 45–50 plants per pot, which were watered three times weekly. When the plants reached heights of between 20 and 30 cm, we collected leaf-bearing stems for experimentation.

Specimens of adult *A. hygrophila* were collected from Changsha, Hunan Province, China, and subsequently bred at the IPP under conditions of 26 ± 1 °C, 75 ± 5% relative humidity, and a 14:10 light:dark photoperiod. The adults used for experimentation had been reared in captivity for 13 generations on potted *A. philoxeroides*. The adult beetles reach maturity after 3 days and, compared with the pre-adult stages, have relatively stronger cold tolerance; we used adult beetles within 3 days of emergence. 

To assess potential differences in the cold tolerance of the different sexes, male and female *A. hygrophila* were separately exposed to low temperatures. Groups of 10 males and 10 females were placed in Petri dishes, with each treatment being performed in triplicate. The criterion used to assess beetle survival was a response to touching the abdomen with a brush. If there was no response, the individual was assumed to have died.

### 2.2. The SCP and FP of Different A. hygrophila Life History Stages

In this study, we determined the SCP and FP of *A. hygrophila* using the thermocouple method with a multichannel SH temperature tester (SH-16; Huaxuan Technology, Shenzhen, China). The measurements were performed in a medical cryogenic storage box (DW−25L262; Haier, China) at −20 °C, in which one end of the thermometer was connected to a thermocouple temperature probe and the other end to a computer. Treatment groups included 10 adult males, 10 adult females, and 10 larvae at each larval instar; each treatment was performed using three replicates. Before measuring the SCP, we initially affixed a thermistor probe to selected *A. hygrophila* specimens using transparent tape, and then placed these in an ultralow temperature refrigerator for measurement. During the measurement process, the thermometer collected data via the corresponding software, which was inputted into the computer. Data were recorded five times per second, and the software automatically generated a temperature change curve. When the insect’s body temperature dropped below 0 °C, as its fluids started to freeze from releasing heat, then the insect’s body temperature suddenly rose, the lowest temperature was recorded as the SCP, and the temperature then suddenly rose to the FP. 

Based on the findings of this experiment, we selected adult beetles to examine cold acclimation, which was designed to approximate the range of identification temperatures recorded in the SCP and FP treatments, and in subsequent experiments, we determined the identification temperatures and times.

### 2.3. Determination of the Identification Temperature of A. hygrophila

The identification temperature refers to the lowest temperature within the sublethal range that an insect can tolerate for a certain length of time. To determine the identification temperature of adult beetles, we assessed their responses to a gradient of temperatures, considering the sublethal temperature between the SCP and FP of these adults. 

To determine the identification temperature of beetles, we subjected these insects to low-temperature treatments within the range between 5 and −14 ± 0.5 °C, starting at 5 °C, and subsequently reducing the temperature in 1 °C steps until reaching −14 °C. Each 1 °C step was considered a treatment. Treatment groups comprised either 10 adult males or females in 9cm-diameter plastic Petri dishes, the bottoms of which were lined with dry filter paper. Each treatment was performed using four replicates. Both water and food were withheld during the experimental period to prevent interference with the influence of freezing on cold tolerance at low temperatures. We also subjected an experimental group to cold acclimation and low temperatures in a biochemical incubator, in which we recorded mortalities of 10% and 80% ± 10% after treatment at −9 °C and −10 °C for 2 h, respectively, and thus considered −10 °C to be the 80% lethal temperature for *A. hygrophila*. As controls, beetles were maintained in a biochemical incubator at 26 ± 1 °C and 75 ± 5% relative humidity under a 14:10 light:dark photoperiod.

### 2.4. Effects of Rapid Cold Hardening on A. hygrophila Viability

To examine the effects of cold acclimation on the cold tolerance of *A. hygrophila*, based on the SCP and FP of *A. hygrophila* adults between −11 and 0 °C, we selected a temperature near the SCP as the identification temperature and a temperature near the FP as the cold acclamation temperature. We also subjected *A. hygrophila* adults to cold acclimation for 1, 2, 3, and 4 h, and accordingly established a 2 h cold acclimation to be the most suitable, as exposure to low temperatures for this length of time did not cause any appreciable physiological damage. Thus, to assess the rapid cold hardening of *A. hygrophila* adults, we examined the responses to a series of temperatures in the sublethal temperature range (−5, −2.5, 0, 2.5, 5, 7.5, and 10 ± 1 °C), to which the beetles were exposed for 2 h. Having thus been cold-hardened, the beetles were subsequently subjected to identification temperature stress for 2 h, after which they were maintained at 26 ± 1 °C to assess the cold hardness and recovery of *A. hygrophila*. As the adults may remain in a state of cold hardness or frostbite during the initial 24 h of the recovery period, mortality was accessed on day 2 after having been returned to 26 °C.

### 2.5. Persistence of Rapid Cold Resistance

As the adults recovered at different temperatures and times after cold acclimation, we established that cold acclimation had a phenotypically plastic effect on the reduction in mortality at the identification temperature. To determine the extent to which the rapidly acquired cold hardness wanes on the return of beetles to a high temperature, we subjected adults reared at 26 °C to low temperatures of 0 and 10 °C for 2 h, after which they were returned to 26 °C for periods ranging from 5 to 60 min extending in 5 min intervals. Thereafter, the beetles that had been allowed to recover for different lengths of time were exposed to a temperature of −10 °C for 2 h, following which they were returned to 26 °C. After a resumption of feeding for 2 days, the mortality of the acclimated beetles was assessed. 

### 2.6. Effects of Different Acclimation Methods on the Cold Tolerance of A. hygrophila

To compare differences in the cold tolerance acquired by adults subjected to rapid cold acclimation, long-term cold acclimation, and fluctuating cold acclimation, we determined the mortalities of *A. hygrophila* cultured at the cold acclimation temperature before exposure to the identification temperature. We used the following four treatments: Treatment 1, Control: adult beetles were maintained in a 26 °C incubator. Treatment 2, Rapid cold hardening: adult beetles were reared at 26 °C for 3 days and then placed in low-temperature incubators at 0 or 5 °C for 2 h. Treatment 3, Long-term cold acclimation: adult beetles were reared at 26 °C for 3 days, and then placed at 15 °C for 24 h. Treatment 4, Fluctuating temperature cold acclimation: adult beetles were initially exposed to −5 °C or 0 °C for 2 h and then gradually cooled to the identification temperature at a rate of 1 °C min^−1^.

### 2.7. Effects of Cold Acclimation on A. hygrophila Longevity

After the rapid cold hardening at 0 or 10 °C for 2 h, the longevity of *A. hygrophila* adults were documented, and there were 20 males and females in each group.

### 2.8. Data Analysis

Before analysis, the data were assessed for normal distribution. The SCP and FP of different beetle instars were analyzed using ANOVA. Repeated-measures ANOVA (LSD, *p* < 0.05) was used to analyze differences in the mortalities of males and females exposed to each temperature and for each time treatment combination. The figures were prepared using GraphPad Prism 6.01 (GraphPad Software Inc., San Diego, CA, USA). *p*-values of 0.05 or lower were considered to be indicative of a significant difference. 

One-way ANOVA (followed by Tukey’s multiple comparison test) was used to determine significant differences between different rapid cold-hardening treatments on survival and SCP using SPSS v. 16.0 (SPSS, Chicago, IL, USA). A two-way ANOVA was used to determine the different cold acclimation responses of males and females.

## 3. Results

### 3.1. The SCP and FP of Different A. hygrophila Life History Stages

The SCP and FP of 1st-instar larvae were approximately −14 and −10 °C, which were established to be the lowest recorded among the different life history stages of *A. hygrophila*. Comparatively, the SCP and FP recorded for the pupae and adults were similar at approximately −10 and −4 °C, with slightly lower values being recorded for the 2nd and 3rd larval instars (Table 1).

### 3.2. Determination of the Identification Temperature of A. hygrophila

We found that all beetles survived when exposed to temperatures of −7 °C or higher for 2 h. When exposed to −8 °C for 2 h, we recorded 20% adult mortality (F7.24 = 175.8, *p* = 0.002), which was statistically different from that of adults reared at 26 °C. The mortalities of adults exposed to −9 °C and −10 °C for 2 h were approximately 30% and 80%, respectively, the latter of which was statistically different from that of adults exposed to −8 °C for 2 h (F7.24 = 175.8, *p* < 0.0001). Based on these findings, we used exposure to a temperature of −10 °C for 2 h as the identification temperature treatment (Figure 1). In addition, for all assessed temperature treatments, we detected no significant differences in the mortalities of adult males and females. 

### 3.3. Effects of Rapid Cold Hardening on A. hygrophila Viability

After rapid cold acclimation at 0, 2.5, 5, 7.5, and 10 °C for 2 h, we detected reductions in the mortalities of *A. hygrophila* adults exposed to the identification temperature. The mortality at identification temperatures was markedly reduced after a 2 h period of rapid cold hardening (Figure 2). However, in terms of mortality, we detected no significant differences among the 2.5 °C-incremental hardening temperatures (F1.14 = 17.20, *p* = 0.1257). There were significant reductions in the mortality of *A. hygrophila* at 0 and 5 °C (F6.14 = 13.09, *p* < 0.0001), and we found that a rapid cold-hardening temperature of 5 °C reduced the mortality of adult females at the identification temperature to a greater extent than that of adult males. These findings thus indicated that cold acclimation treatments at 0 and 10 °C for 2 h were the most effective in enhancing the cold resistance of *A. hygrophila*. Following exposure to a temperature of 0 °C for 2 h, we detected a reduction in adult mortality to 36.7% (F6.14 = 13.09, *p* < 0.001), and mortality at the identification temperature was also lower in beetles exposed to 5 °C or 10 °C (F6.14 = 13.09, *p* < 0.001). Exposure to 5 °C for 2 h was found to result in an approximate 50% reduction in female mortality at the identification temperature (F6.14 = 13.09, *p* < 0.001), whereas pre-exposure to temperatures of 0 and 5 °C considerably enhanced the cold resistance of *A. hygrophila* when adult beetles were exposed to low temperatures for 2 h. However, having been subjected to cold acclimation at −5 °C, we recorded a notable reduction in the survival of *A. hygrophila* when exposed to the identification temperature for 2 h (F6.14 = 0.015, *p* = 0.9087). 

### 3.4. Persistence of Rapid Cold Resistance

We found that after 15 min of having returned beetles to 26 °C following rapid cold acclimation, the acquired cold tolerance of *A. hygrophila* returned to pre-acclimation levels. Following cold acclimation treatments, we transferred adults to the normal rearing temperature of 26 °C for periods ranging from 5 to 60 min at 5 min intervals. Having allowed recovery at 26 °C for these different durations, the beetles were exposed to −10 °C for 2 h, and we accordingly found that recovery at 26 °C for a period of 15 min or longer essentially negated the effects of cold acclimation in reducing mortality at the identification temperature (Figure 3). The highest mortality was recorded following cold hardening at 0 °C for 2 h (36.67%), although no significant differences were detected between those beetles allowed to recover for 15 and 30 min (F2.3 = 0.222, *p* = 0.813). Contrastingly, the mortality of beetles allowed to recovery for 15 min following exposure to 0 °C for 2 h was found to differ significantly from that of untreated control beetles (F10.22 = 6.804, *p* < 0.001). However, the cold tolerance of adults acquired by rapid cold hardening at 0 °C for 2 h was soon lost within 30 min on the return of adults to 26 °C. Accordingly, the degree of mortality occurring at the identification temperature differs depending on the length of post-cold hardening recovery. 

### 3.5. Effects of Different Acclimation Methods on the Cold Tolerance of A. hygrophila

Although we established that exposure to 15 °C for 24 h contributed to reducing the mortality of female *A. hygrophila* at the identification temperature, the mortalities did not differ significantly from those recorded for the control group (F4.2 = 3.000, *p* = 0.643). In contrast, the mortality of males, which had been reduced from 83.33% to 40.00%, was found to differ significantly from the level of control group mortality (F4.2 = 6.000, *p* = 0.0147).

Rapid cold hardening and fluctuating-temperature cold acclimation had different effects depending on temperature and duration. When we initiated low-temperature treatment at −5 °C, we recorded a reduction in the mortality of beetles at the identification temperature, with the resulting mortality being found to differ significantly from that of control group beetles (F2.12 = 6.485, *p* = 0.048). Contrastingly, we detected no significant differences in mortality following rapid cold hardening at −5 °C for 2 h (F2.12 = 0.516, *p* = 0.512). When commencing the low-temperature treatment at 0 °C, both rapid and fluctuating-temperature cold acclimation effectively reduced adult mortality at the identification temperature, with the effect obtained using rapid cold acclimation being superior to that obtained in response to fluctuating-temperature cold acclimation (Figure 4). We found that the mortality of adults cooled from 0 °C at a rate of 1 °C min^−1^ was similar to that of the control adults, which experienced no cold treatment and were not exposed to the identification temperature, whereas mortality was significantly reduced in response to rapid cold hardening at 0 °C for 2 h (F6.14 = 13.09, *p* < 0.001).

### 3.6. Effects of Cold Acclimation on A. hygrophila Longevity

Exposure to the identification temperature contributed to a clear decline in the longevity of males and females, as a consequence of suffering irreversible cold-induced damage (Figure 5). Furthermore, we detected differences in the average lifespans of adult males and females, with the latter being characterized by superior cold tolerance. 

We also established that the longevity of adults that were cold-acclimated at 0 °C was notably higher than that of beetles acclimated at other temperatures, with mortality on the third day of cold exposure being considerably lower than that at any other acclimation temperature. 

## 4. Discussion

In this study, we investigated the mortality of *A. hygrophila* adults when subjected to the identification temperature following rapid cold acclimation, long-term cold acclimation, and fluctuating-temperature cold acclimation. Our findings indicated that the mortality of adult *A. hygrophila* increases with a decline in temperature and prolongation of exposure time. Furthermore, when subjected to different acclimation procedures, we detected significant reductions in the mortality of *A. hygrophila* adults at the identification temperature and found that the protective effect of cold acclimation can readily wane when the beetles are returned to a temperature of 26 °C. In subsequent analyses to evaluate the effects of cold acclimation on *A. hygrophila* adults, we assessed the longevity of these beetles, which provided evidence to indicate that cold acclimation may enhance the cold tolerance of these beetles.

The SCP can be used to measure the response of natural populations of animals to climate change [27]. In this study, we selected the SCP and FP as parameters for determining the cold tolerance of target insects, thereby providing a representative coverage of the physiological and biological aspects of the responses to low temperatures. The determination of these indicators can serve as reference values for the subsequent low-temperature acclimation of a species. In this regard, the conditions that induce rapid cold hardening may differ for freeze-intolerant and freeze-tolerant insect species [6]. Although freeze-intolerant species may survive to the FP, a certain proportion of the population is killed or fatally injured before freezing. The occurrence of and change in this mortality becomes obvious at the identification temperature. Whereas cold acclimation is readily detected in freeze-intolerant insects and cold-intolerant insects, *A. hygrophila* is a cold-susceptible insect, which is a phenotype between cold-tolerant and cold-intolerant insects [28,29]. In the present study, we established that the sub-lethal low-temperature zone for *A. hygrophila* adults determined by the SCP and FP coincides with an approximate temperature range of −10 to 0 °C, and accordingly selected temperatures within the sub-lethal range to conduct our cold acclimation experiments. 

The physiological regulation of tolerance to extreme temperatures may differ depending on whether a species is active in summer or winter [30]. In the present study, we determined the mortality, longevity, and other physiological indicators of cold acclimation of *A. hygrophila* during winter. Given that the cold tolerance of insects may undergo seasonal changes, it is conceivable that the cold tolerance of *A. hygrophila* adults in summer differs from that characterized in this study. The cold tolerance of *A. hygrophila* may also differ according to geographical population, rearing procedures, and age. Thus, as biological indicators are likely to be influenced by changes in the microenvironment, follow-up studies should be conducted to determine the cold tolerance of these insects in other seasons to gain a more comprehensive understanding of their survival potential. 

When exposed to a gradient of cold temperatures, we detected a significant reduction in the mortalities of *A. hygrophila* adult males and females. At temperatures lower than −7 °C, there was a marked increase in mortality, with 80% adult mortality being recorded among those beetles exposed to −10 °C for 2 h. This indicates that there was a significant reduction in the mortality of adults. However, when exposed to 0 or 10 °C for 2 h before exposed to −10 °C for 2 h, adult survival was significantly adversely affected at temperatures within the vicinity of −10 °C. Exposure to 0 or 10 °C was found to induce rapid cold acclimation, with the latter of these two temperatures promoting a more pronounced effect. In this regard, although the findings of some studies have indicated that a 30 min exposure to temperatures within the sublethal temperature region is sufficient to induce cold acclimation [31], in the present study, we established that a 2 h exposure at 0 or 10 °C was necessary to promote a significant increase in cold resistance of the adult males and females of *A. hygrophila*. 

Cold acclimation is a physiological strategy that is widely adopted by ectotherms to adapt to sudden changes in temperature and has been documented in numerous insects within a temperature range of between 0 and 10 °C [31,32,33,34,35]. Among coleopteran species, it has been established that in several species, a period of cold acclimation can contribute to enhancing their cold tolerance [36,37,38], and studies that examine plastic responses such as cold acclimation will contribute to predicting the effects of climate change on species distribution and survival [39]. In the present study, we found that both the males and females of *A. hygrophila* undergo cold acclimation and that individuals thus acclimated are characterized by significant reductions in mortality on subsequent exposure to low temperatures.

In many insects, a reduction in cold tolerance coincides with an increase in the post-acclimation recovery time, with the protective effect of cold acclimation disappearing after a certain threshold of recovery, as has been observed in *Corythucha ciliata* [9], *Liriomyza trifolii*, and *Liriomyza sativae* [40]. Consistently, in the present study, we found that unless adults of *A. hygrophila* are maintained at low temperatures, the effects of cold acclimation may disappear, as indicated by our finding that beetles reverted to a cold-susceptible phenotype within 15 min of recovery at 26 °C.

Rapid cold acclimation is now believed to be a common phenomenon in many insects, in which it plays an ecologically important role in survival [6]. Appropriate cold treatment does not affect the developmental duration and offspring fitness [41]. We suspect that this type of acclimation might also benefit *A. hygrophila* adults and their offspring under natural conditions. In the winter in northern and eastern China, temperatures may drop to below 0°C in the colder regions. Cold acclimation can contribute to reducing the mortality of *A. hygrophila* during the overwintering and early spring seasons, which has ecological relevance for its survival.

In this study, we found exposure to certain temperatures for defined lengths of time can enhance the survival of *A. hygrophila* adults at the identification temperature, although we also established that their longevity was affected by constant exposure to low temperatures. Freezing can also cause mechanical damage to cells and tissues, as a consequence of dehydration or metabolic perturbation [42]. Given that the identification temperature lies within the range of low temperatures that are sublethal to *A. hygrophila* adults, beetles subjected to a preliminary rapid cold acclimation treatment would predictably be detrimentally affected by a sudden increase in temperature. However, in this regard, we detected a difference in the longevity of adult males and females, with the latter surviving longer than the former, indicating that *A. hygrophila* females may be physiologically better adapted to survive when exposed to low temperatures with the accumulation of antifreeze physiological substances.

Studying insect cold resistance entails assessing the survival of individuals subjected to different combinations of cooling rates, exposure periods, and minimum temperatures under laboratory conditions [5], and in this regard, it has been found that there are no significant differences in the cold tolerances of field-collected and laboratory-reared second-generation populations [43]. Except for treatment temperature, control group beetles were reared in identical conditions to the cold-acclimated beetles. In addition, comparisons of laboratory and field survival can provide an estimate of winter mortality that may be caused by factors other than low temperatures [44,45]. The cold acclimation of *A. hygrophila* adults may enhance their survival and overwintering potential in certain regions of northern China, which would have important implications for controlling the northward spread of *A. philoxeroides* in China.

The distribution of *A. hygrophila* is determined by several factors relating to climate, including survival at low temperatures and overwintering capacity [46]. When initially introduced in China, populations of *A. hygrophila* were unsystematically released in several spatially dispersed areas, although they subsequently became established in Yunnan, Hunan, Fujian, and Hubei provinces and Chongqing City, whereas populations failed to become established in Hebei, Beijing, and other northern regions of China [15]. In addition, there were no overwintering populations found in Jingzhou City in Hubei province [47]. With the global warming, the lowest temperature in China may increase, so the increase of cold tolerance may help insects in successful wintering. If by subjecting *A. hygrophila* adults to a period of cold acclimation we can enhance the cold tolerance of these beetles, thereby reducing mortality when exposed to lower ambient temperatures and increasing the probability of overwintering in colder areas, we believe it would be feasible to establish populations in areas that they have hitherto failed to colonize, which would contribute to reducing the cost of *A. philoxeroides* control.

## 5. Conclusions

In this study, we established that adults of the chrysomelid beetle *A. hygrophila* undergo rapid cold hardening in response to short-term exposure to low temperatures of 0 or 10 °C for 2 h, 15 °C for 24 h, or a gradual cooling from 0 to −10 °C at 1 °C min^−1^. However, the cold tolerance thus acquired tends to gradually wane on the return of these beetles to a rearing temperature of 26 °C. In the context of climate warming, rapid cold hardening may enable *A. hygrophila* adults to overwinter in the warmer regions of northern China.

## Figures and Tables

**Figure 1 insects-14-00058-f001:**
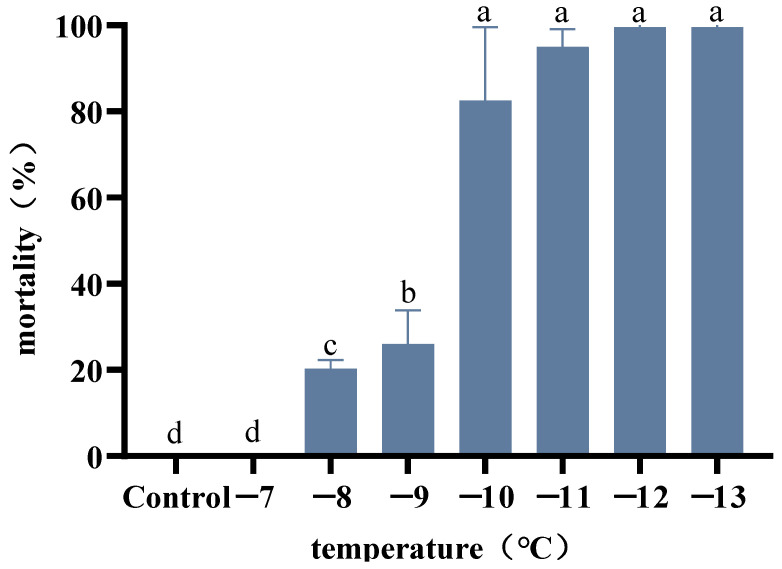
Mortality of the adult males and females of *Agasicles hygrophila* after exposure to identification temperatures for 2 h. Adults maintained at 26 °C served as untreated controls. Different letters refer significant differences in the mortality of beetles exposed to the different temperatures for 2 h (Tukey multiple comparison test at *p* < 0.05). Based on these results, we used exposure to a temperature of −10 °C for 2 h as the identification temperature treatment. All values are presented as the mean ± SE.

**Figure 2 insects-14-00058-f002:**
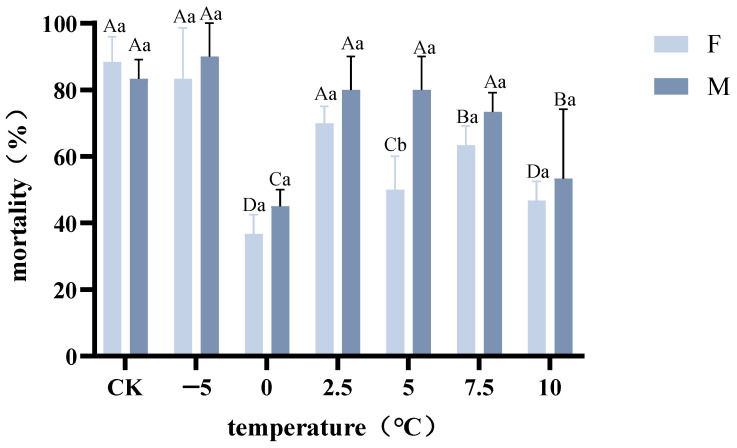
Effects of rapid cold hardening on the mortality of adult *Agasicles hygrophila*. We used repeated-measures ANOVA, followed by the LSD test (*p* < 0.05). Capital letters refer to the differences in the mortality of *A. hygrophila* following treatment with the same temperature for the different sexes. Lowercase letters refer to the differences in the mortality of *A. hygrophila* following treatment with different temperatures for the same gender of different temperature RCH. All values are presented as the mean ± SE.

**Figure 3 insects-14-00058-f003:**
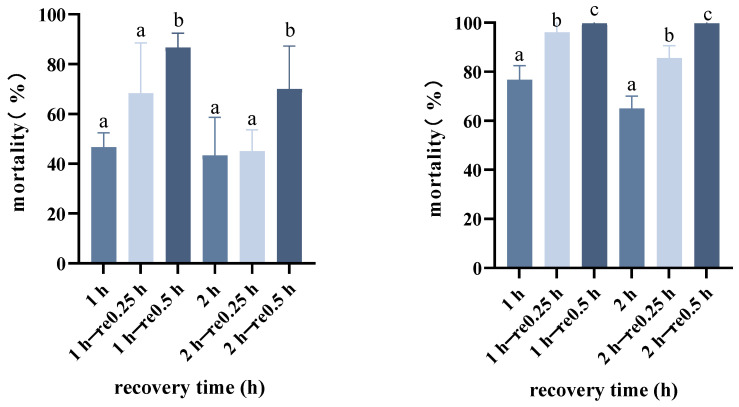
Mortality of *Agasicles hygrophila* adults exposed to low temperature and subsequent recovery for different lengths of time. Different letters refer significant differences in the mortality of beetles following different recovery times and were analyzed by the LSD test (*p* < 0.05). 1 h and 2 h are the treatment times, re −0.25 h and 0.5 h are the recovery times at 26 °C. All values are presented as the mean ± SE.

**Figure 4 insects-14-00058-f004:**
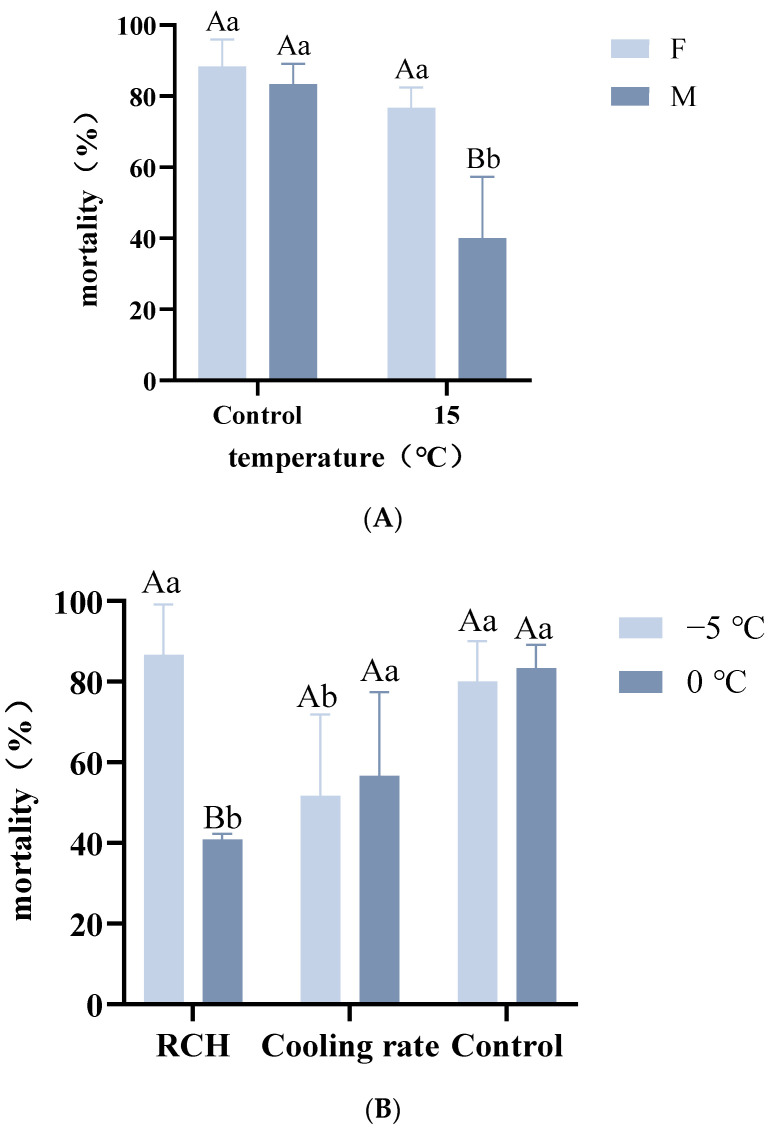
The mortality of *Agasicles hygrophila* adults acclimated using different procedures. (**A**) Mortality of *A. hygrophila* adults at the identification temperature after long-term acclimation compared with that of untreated controls. (**B**) Mortality of *A. hygrophila* adults after rapid cold hardening (RCH) and gradual cooling at 0 and 5 °C. Control insects were directly exposed to −10 °C for 2 h without pre-exposure cold acclimation. Capital letters refer to the differences in the mortality of *A. hygrophila* following treatment with the same temperature for different genders. Lowercase letters refer to the differences in the mortality of *A. hygrophila* following treatment with different temperatures for the same gender. Differences in mortality were analyzed using repeated-measures ANOVA, followed by the LSD test (*p* < 0.05). All values are presented as the mean ± SE.

**Figure 5 insects-14-00058-f005:**
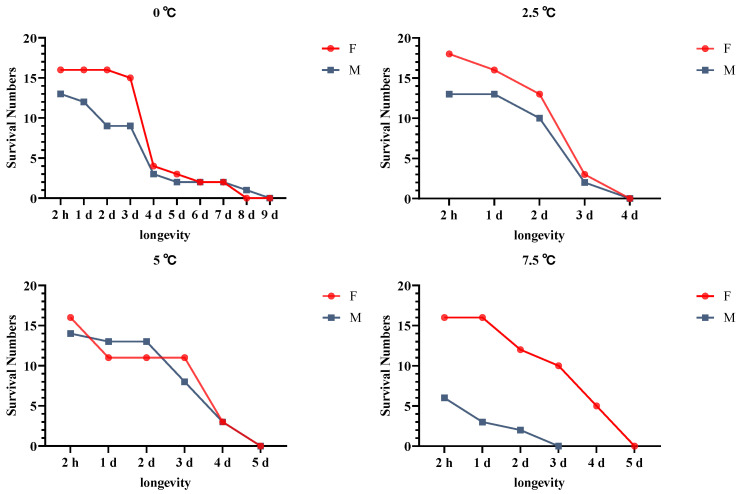
The longevity of *Agasicles hygrophila* adults after rapid cold-hardening and discrimination temperature treatment. Differences in the longevity of adults after different periods of recovery were analyzed using a repeated-measures ANOVA, followed by the LSD test (*p* < 0.05). All values are presented as the mean ± SE.

**Table 1 insects-14-00058-t001:** Supercooling point (SCP) and freezing point (FP) for different life history stages of *Agasicles hygrophila*. All values are presented as the mean ± standard error (SE). Values are in degrees Celsius.

Developmental Stage	SCP	FP
1st instar	–14.54 ± 4.06	–10.33 ± 5.22
2nd instar	–11.77 ± 1.96	–4.69 ± 2.51
3rd instar	–10.67 ± 2.93	–2.68 ± 2.19
Pupa	–10.85 ± 3.89	–1.92 ± 1.41
Adult female	–9.81 ± 2.06	–4.19 ± 2.61
Adult male	–8.38 ± 2.85	–3.12 ± 2.35

## Data Availability

The data presented in this study are available on request from the corresponding author.

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
