# Peer review of "Cold Acclimation and Supercooling Capacity of Agasicles hygrophila Adults"

_insects, 2023, doi:10.3390/insects14010058_

Round 1
Reviewer 1 Report
In this study, the authors compared the rapid cold acclimation ability of 3-day-old Agasicles hygrophila adults and detected their supercooling points (SCPs) by cold and freezing injuries. They compared and evaluated the rapid cold-hardening ability of A. hygrophila adults, which can avoid low sublethal temperatures and sudden temperature changes in their normal habitats. In general, proper scientific methods were used, and the results are of interest. The data is sufficient to support its main conclusion and the study matches the scope of insects journal. However, the manuscript needs careful proofreading and revision. Grammar mistakes are undermining the significance of this study. For example check the title of section “2.3. Determination of identification Temperature of A. hygrophila” Therefore, the authors should submit it to the editing company for grammar modification. I would like to recommend it to be published with suggestion of some major points listed below, which should be addressed in this major revision.
-L21-25: The abstract include a lot of background information. It should be rewrite completely. I suggest to summarize the mentioned sentences and try to add key results in abstract.
-The introduction section is not coherent, included a lot of general information. The authors should revised the introduction section entirely especially the first paragraph. Try to remove the general sentences. Discuss the main points in details, it should be to the point and specific. No need to add general information.
-L115-116: “We used newly emerged adult beetles within 3 days for the experiment” WHY? This point should be discussed here. Why you choose 3 days old adults?
-L166-168: “To explore the effects of cold acclimation on the cold tolerance of A. hygrophila, we set a series of temperatures, including –5, –2.5, 0, 2.5, 5, 7.5, and 10 ±1 ℃, for rapid cold hardening for 2 h in the sublethal temperature range of A. hygrophila adults.” Okay, but why you select specifically –5, –2.5, 0, 2.5, 5, 7.5, and 10 ±1 ℃? And why 2h exposure? The authors should discussed these points in materal and methods section. The authors should provide each and every detail of the methodology.
-The authors should add complete statistical values in results section.
-Figure 1: No need of different colors. The authors should reconstruct figures to make it more striking. Here different colors looks not okay. May be authors should keep one color for all bars.
-Figure 3: The authors missed the asterisks. Please check it.
-L112: CAA to “CAAS”
-Again, the English of this manuscript is very week. The authors should revise it by professionals or English editing company.
Author Response
Dear Editor,
Thank you very much for your letter and advice on our manuscript. We have resubmitted a revised version of the manuscript in accordance with the recommendations of the technical editor. We have addressed the comments raised by the reviewer. Please see the attachment. We hope that the revision is acceptable and look forward to hearing from you soon.
With best wishes.

Reviewer 2 Report
This manuscript’s purpose, as outlined in the Introduction, was to assess the cold tolerance of Agasicles hygrophila, a biological control agent of Alternanthera philoxeroides, for potential northward expansion to determine if it can continue to be an effective biocontrol agent in colder climates. The series of experiments conducted provide some interesting and useful information on the A. hygrophila cold tolerance, but the authors never return to their original question of where it can survive northward expansion in China. Also, the manuscript and presentation of results is very confusing and difficult to follow, and there are many statements in the discussion that are not supported by the data. While there is useful data generated, it is my opinion that it will need to be extensively rewritten to improve the organization, clarity, and focus of discussion.
INTRODUCTION:
Lines 53-59: It is not necessary to provide the specific details of the study with E. finlandicus or C. ciliate. These references can be cited as examples of phenotypic plasticity in the response to cold temperatures. Plus, it’s not clear what “0.4 °C/min” refers to - such detail not needed.
Lines 60-73: Some more information is needed on the type of plant and status of Alternanthera philoxeroides. What family does it below to? Similarly, what is A. hygrophila – what family does it belong to, and where is it native to?
Lines 74-82: much of this paragraph can be deleted. All that seems to be necessary, because all that this paragraph says is stated in the last sentence. But, considering this manuscript did not compare cold tolerance of different life stages, it is not necessary.
Line 87-89, and 92-94: These sentences add nothing, so delete.
Paragraph starting at Line 96: First, replace detected with estimated. Also, there are some key terms presented here for the first time that should be defined, such as supercooling point, rapid cold-hardening, and sublethal temperature. On line 100, what “threshold” is being estimated, and the “previous models” should be referenced.
General comments on paragraph beginning on line 96
MATERIALS AND METHODS:
Line 113: what is meant by “cut.”
Line 117: It should read “In these experiments…” Also, I find the term “treatment” to be somewhat confusing. While technically these are different treatments, it’s easier to follow by stating insects were exposed to different temperatures.
Line 123: It’s not clear the difference between supercooling point and freezing point.
Line 138: This description is confusing; what is meant by “cold domestication?” domestication implies taming something or cultivating it, and is an anthropomorphic term. I think acclimation is the correct term here.
Also, here and in the next paragraph (beginning line 146), what are the definitions of identification temperature and identification time? And how does this differ from “recognition temperature” introduced in the line 146. These terms seem to be used interchangeably.
Line 170: Again, “coma” is an anthropomorphic term not really not appropriate here. Also, how long were beetles kept at room temperature before final mortality was recorded? This brings up another area of confusion, was this same procedure following for determining mortality – bringing adults back to room temperature for a specified time? This was not described for the previous experiment.
Also, what is “room” temperature referenced here and in several other places. If this is 26C, it should be 26C and not room temperature.
Line 186: “We conducted normal feeding…” needs to be rewritten.
Line 189 paragraph: Description of these treatments is confusing. What is meant by “…cultured at the recognition temperature before exposure to the recognition temperature.” I don’t know what this means, nor do I follow the description of the 4 treatments. Maybe a figure depicting what was done would help.
RESULTS:
Section 3.1: The materials and methods only described experiments with adults, so how come these different life stages are listed in Table 1? IF this is going to be included, the methodology for how this study was done for all stages should be described.
Line 222 and throughout: not substantially is not a term to use in a scientific publication. We are interested in values that are statistically or not statistically different.
All Figures: Indicate what the asterisks mean. Does this signify that treatments were different from the control, and if so what does 1 vs. 4 stars mean.
Section 3.3: Where are results from the variable temperature acclimations described in the paragraph starting on line 172?
Section 3.4: This is section is quite confusing, and Fig. 3 needs much more explanation. I have no idea what abbreviations below columns mean. Also, I take that there were no significant differences since there are no asterisks above columns. If so, much of this can be condensed.
Section 3.5: What is RCH (first mention of this term). If these results correspond to section 2.6 in Methods, it’s hard to tell they are even connected. 4 treatments were described in the methods, but there’s no evidence of results from four treatments in Fig. 4.
Section 3.6: This section is not even described in the Material and Methods. Where did this data come from?
DISCUSSION: There are several references to climate change that have no relevance to this study – it seems to be thrown in just to state the term climate change. Also, in the introduction it is implied that the reason for this study was to determine if Agasicles hygrophila has the capability to expand northward with its host. This is never discussed.
Line 317 Paragraph: These terms should all have been defined earlier in the manuscript. But, many were never mentioned before this, so why even bring them up?
Line 333: It’s hard to think scientists considered rapid cold acclimation as ecologically irrelevant? Is there a reference to substantiate this statement?
Line 350: Reproduction and offspring were never evaluated in this study, so why is this even mentioned? Should be deleted.
Line 354 Paragraph: Not sure the relevance of this first sentence to this study. Also line 359 is written as if this is a universal occurrence among arthropods, which is not justified.
Line 370: What does this refer to? There was no comparison of lab and field survival morality in this study.
Line 383: This study did not measure fecundity.
Author Response

(The authors gave the same response as above.)

Reviewer 3 Report
I have read the manuscript entitled "Cold Acclimation and Supercooling Capacity of Agasicles hygrophila Adults" submitted to Insects. The authors examined the cold tolerance of a beetle species in different ways: fast hardening, longer acclimation, differences among life stages and between the sexes, and longer-term effects on survival. The authors invested much effort, and it is clear they did a lot of work. That said, the manuscript is absolutely unclear. The Abstract does not describe well the paper's content. The Results do not fully match the Methods, the Discussion is poorly written and most of it does not discuss the Results. The same terms are referred to in more than a single way, adding to my confusion. There are no clear statistics in the paper (but sometimes only in the figures). It is therefore hard to judge whether the manuscript can be published at the end, not before a very thorough revision is done to improve the manuscript's clarity. Please see specific comments below.
L 40: Better define what rapid cold hardening is. See for example Bowler 2005 Journal of Thermal Biology 30:125-130.
Line 43: A reference or two is needed here.
L 47-49: How do you know that the differences among populations are only plastic? Why cannot they be genetic? This sentence seems wrong to me.
L 50: Why "feeding" and not "keeping"? The flies were kept under such temperatures, and were not only fed there, right? L 54: Again, why "feeding conditions"?
First paragraph of the Introduction: I would like to read here also about possible mechanisms of cold hardening. How do insects do this?
L 60: When you first mention Alternanthera philoxeroides, please write what it is. It is unclear from the text, and I had to google it. Later, you can use the scientific name. L 64: Same here, write which insect it is.
L 66: Remove basic.
L 74: What is cold domestication and how does it differ from cold hardening or acclimation? Please define the terms you use and do not use two terms to describe the same phenomenon.
Third paragraph of the Introduction: The transition from the second to the third paragraph is not smooth. The first paragraph discusses cold tolerance in general. The second focuses on the studied species in China. The third is general and once more describes cold tolerance and acclimation. The Introduction is therefore not well organized. Finish discussing the general topics first before you move to describe your studied system and its importance.
L 75-77: If rapid cold hardening has been described in many species then better provide more than a single reference.
L 79-82: It is unclear why here you switch to discuss differences between adults and juveniles in cold tolerance. You jump between different topics and describe each quite briefly.
The fourth paragraph of the Introduction changes the topic again. Here, there is a brief switch to the climate in China, and then returning to cold acclimation. The Introduction is not well structured.
L 95: On the one hand, "we compared the rapid cold acclimation ability of 3-day-old A. hygrophila adults" is not all that was done here. For example, you also examined the persistence of rapid cold resistance. I think you should better describe here what you did. On the other hand, you did not compare juveniles and adults. So why do you mention such a comparison in the Introduction before?
L 96: "Supercooling points": Define it when it first appears.
End of Introduction: What did you expect to find?
L 113: Is it the optimal temperature for this insect?
L 117: What do you mean by "separately treated"? Can you be more specific?
L 123-127: The difference between SCP and FP is not very clear.
L 128: Repeat here the motivation to measure SCP and FP. Why prefer these two measures over other cold tolerance measures?
129: What is SH here?
L 132: Cooling point = SPC?
L 133: What do you mean when writing "live beetles"? Were many of them dead?
L 138-140: I don't get this sentence. The temperature dropped and the increased fast? But you are in control of the temperature? Or are you measuring the insect's body temperature? This is unclear for someone not familiar with your specific methods.
L 140: 15 of each sex?
L 145-146: In the title, it is written "identification temperature", but a sentence later you write "recognition temperature". Is it the same? Use consistent terminology. Also, if it is a common term, please cite other papers that used the term as you define it here.
L 153: How many individuals of each sex per treatment?
Lines 174-176: First, you wrote that "Here, we compared the survival rates of 174 the beetles after treatment for 2 h, 1 d, and 2 d", but then "We counted the dead individuals daily until the last individual died". Please explain.
L 183: What is "temperature-recognition mortality"?
L 185: Why 2 h?
L 187: For how long?
What is the difference between "Rapid cold hardness" in L 193 and "rapid cold acclimation" in L 189? If they are the same, it is important to use consistent terminology. The current presentation is confusing.
L 195: "at 0 or 5 °C for 1 or 2 h, respectively": Why? What is the rationale of this treatment combination?
L 196: "and then placed them at 15 °C in the incubator": For how long?
L 200, data analysis: Please be more specific and write for each test what were the response and explanatory variables, as well as the test you used.
L 207-211: Only here, I got it you examined different life stages. Was it written before in the Methods? I found it unclear. Also, where are the statistical results (P values)?
Title of the Results and Methods sections: Your titles in both sections are not identical, making it difficult to link the description of each sub-experiment and its result. I recommend identical titles.
Table 1: Values are in Celsius degrees?
L 219-228, L 237-246, L 254-272: Also here, statistics (test statistics and p values) are missing.
L 232: Tukey instead of Turkey.
L 244-246, L 264-265, L 271-272: This is an interpretation of the results and belongs in the Discussion.
Figure 2, Figure 4: I don't get the significance signs. What do you compare with what?
Figure 3: Where can one see the paired comparisons you did? What do the signs on the x axis mean?
L 317-325: This paragraph is out of the paper's context. It is recommended to write a few words on how to identify cold tolerance, but not here. Instead, the first paragraph of the Discussion should present the main findings and their importance.
L 326-332: The second paragraph also does not fit. It is not directly related to your results. Only in L 335 there is something meaningful written, directly related to the presented experiments.
L 352-353: Please explain and better justify how global warming is related to your study. Unclear if it is really expected to affect the studied species.
L 354-358: Unclear what you wish to write here. For example, what is "cold tolerance index"?
L 359-363: How do these sentences refer to your study? Unclear. The whole paragraph does not discuss the results.
L 376: "The newly hatched larvae, being small, require less energy, resulting in a lower SCP": How do you know that being small and requiring less energy leads to lower SCP?
Author Response

(The authors gave the same response as above.)

Round 2
Reviewer 1 Report
The authors have addressed all comments with full justifications. I recommend this manuscript for publication.
Author Response
Dear Editor,
Thank you very much for your letter . We have resubmitted a revised version of the manuscript in accordance with the recommendations of the technical editor. We have addressed the comments raised by another reviewer, and we have revised the article slightly.
With best wishes.
Reviewer 3 Report
This is the second time I read this manuscript. The manuscript has greatly improved. That said, there are some minor editing problems, which I highlight below. After these problems are solved, the manuscript can be accepted for publication, in my opinion.
105-106: An unclear sentence.
110-111: It is still unclear and undefined what "identification temperature" is.
113: Is rapidly acquired cold resistance = hardening? You still have many terms, and I recommend making an effort not to refer to the same thing in more than a single term.
135: Cold tolerance. For experimentation…
160, 214: acclimation instead of acclamation. Rapid cold acclimation = hardening? Please use consistent terminology…
221: longevity was measured (or documented). Species name in italics.
222: Unclear what you mean to say in "the females and males were measured respectively".
233: In two-way ANOVAs there are two independent variables. Here, you examine the effect of sex on cold acclimation responses. What was the second independent variable?
Table 1: Perhaps write "Adult female" and "Adult male" to make it clearer.
Figure 1: Are means and SE presented here?
264: Full stop after temperature.
279: Define in the main text what the recognition temperature is.
281: I think there is a mistake here in the numbers after the ANOVA F: "F 1, 4". Please check.
286: You sometimes use the word "gender" and sometimes "sex". I would stick to one, preferably sex.
292: "attributable to the fact" – to which fact?
Figure 3: Sexes are presented together? Write in the figure captions what the x labels stand for.
Figure 4: Write in the figure captions what CK stands for.
Figure 5: Are the initial numbers of females and males the same? If no, then better present proportions or percentages, as you did in the previous figures. If yes, you can keep the figure as it is now.
365: Remind the readers what the recognition temperature is.
368: Unclear what "treatment time" is. Do you mean exposure to a certain temperature?
400-402: Remove. This sentence is uninformative.
434: What is an appropriate cold treatment? Is it species specific? If so, does it refer to the studied species?
The sentence in 450-451 is not an explanation of 449-450 but just a repetition in other words. What can be the mechanism or the reason behind this result?
459-461: "Bale (1987) [6] found that temperature is an important factor affecting insect cold tolerance, and therefore low-temperature tolerance should be investigated under laboratory conditions": I don't get this sentence. What is the link between an investigation under laboratory conditions and this finding? What do you intend to express here?
468-472: Vague and not directly related to the current study. I would remove.
The paragraphs in lines 473-486 and lines 487-494 are quite similar with repeating content. I would merge them, remove overlapping sentences, and keep everything shorter.
Author Response
Dear Editor,
Thank you very much for your letter and advice on our manuscript. We have resubmitted a revised version of the manuscript in accordance with the recommendations of the technical editor. We have addressed the comments raised by the reviewer. Please see the attachment.
We are looking forward to hearing from you.
With best wishes.
